# Death and Disability Reported with Cases of Vaccine Anaphylaxis Stratified by Administration Setting: An Analysis of the Vaccine Adverse Event Reporting System from 2017 to 2022

**DOI:** 10.3390/vaccines11020276

**Published:** 2023-01-28

**Authors:** Rachel C. Klosko, Sarah E. Lynch, Danielle L. Cabral, Kanneboyina Nagaraju, Yvonne A. Johnston, Joshua D. Steinberg, Kenneth L. McCall

**Affiliations:** 1School of Pharmacy & Pharmaceutical Sciences, Binghamton University, Johnson City, NY 13790, USA; 2Master of Public Health Program, Decker College of Nursing and Health Sciences, Binghamton University, Johnson City, NY 13790, USA; 3United Health Services Family Medicine Residency and Upstate Medical University College of Medicine Clinical Campus, Johnson City, NY 13790, USA

**Keywords:** anaphylaxis, vaccine administration, adverse event, administration setting, COVID-19 vaccination, influenza vaccination

## Abstract

The serious nature of post-vaccination anaphylaxis requires healthcare professionals to be adequately trained to respond to these hypersensitivity emergencies. The aim of this study was to compare outcomes reported with cases of vaccine anaphylaxis stratified by administration setting. We queried reports in the Vaccine Adverse Event Reporting System (VAERS) database from 2017 to 2022 and identified cases involving anaphylaxis with an onset within one day of vaccine administration. The primary outcome was the combined prevalence of death or disability for each setting while the secondary outcome was the prevalence of hospitalization. Adjusted (age, sex, prior history of allergy, vaccine type) odds ratios (aOR) and associated 95% confidence intervals (CI) were calculated using logistic regression analysis. A total of 2041 cases of anaphylaxis comprised the primary study cohort with representation in the sample from all 50 US states and the District of Columbia. The mean age was 43.3 ± 17.5 years, and most cases involved women (79.9%). Cases of anaphylaxis were reported after receiving a coronavirus vaccine (85.2%), influenza vaccine (5.9%), tetanus vaccine (2.2%), zoster vaccine (1.6%), measles vaccine (0.7%), and other vaccine (4.5%). Outcomes associated with reports of vaccine anaphylaxis included 35 cases of death and disability and 219 hospitalizations. Compared with all other settings, the aOR of death and disability when anaphylaxis occurred was 1.92 (95% CI, 0.86–4.54) in a medical provider’s office, 0.85 (95% CI, 0.26–2.43) in a pharmacy and 1.01 (95% CI, 0.15–3.94) in a public health clinic. Compared with all other settings, the aOR of hospitalization when anaphylaxis occurred was 1.02 (95% CI, 0.71–1.47) in a medical provider’s office, 1.06 (95% CI, 0.72–1.54) in a pharmacy, and 1.12 (95% CI, 0.61–1.93) in a public health clinic. An analysis of a national database across six years revealed no significant differences in the odds of death/disability and odds of hospitalization associated with post-vaccination anaphylaxis in the medical office, pharmacy, and public health clinic compared with all other settings. This study expands our understanding of the safety of immunization services and reinforces that all settings must be prepared to respond to such an emergency.

## 1. Introduction

Vaccines are essential public health tools with a favorable benefit/risk profile at the individual and population levels [1]. Vaccines have routinely increased individual and population-based immunity and subsequently reduced death, morbidity and hospitalization from infectious diseases [2,3]. Anaphylaxis after vaccination is rare and can be due to a reaction to any of the vaccine components such as antigens, adjuvants, or inactive ingredients [1,4]. Post-vaccination anaphylaxis is a life-threatening, multi-organ (cutaneous, gastrointestinal, respiratory and cardiovascular systems) adverse event with onset typically within minutes [1,4]. Anaphylactic reactions are triggered by the binding of the allergen to immunoglobulin E (IgE), whereas anaphylactoid reactions (which are clinically indistinguishable) are triggered by mast cell activation independent of IgE [1]. Risk of anaphylaxis after influenza vaccination is estimated to be approximately one case per million doses [4]. The incidence of anaphylaxis after coronavirus (COVID-19) vaccination has been estimated to be between five and ten cases per million doses [5,6,7]. Excipients in both the mRNA and adenovirus vector-based COVID-19 vaccines have been found to be allergenic [8]. The serious nature of this adverse event requires healthcare professionals to be adequately trained to respond to post-vaccination hypersensitivity emergencies including anaphylaxis [8].

Anaphylaxis following vaccination is not always predictable or preventable. Consequently, any professional administering a vaccine must be prepared to manage an anaphylactic reaction [9]. According to the CDC, this preparation includes training to screen persons with vaccine contraindications or precautions, to implement appropriate post-vaccination observation periods, to identify signs and symptoms of anaphylaxis, to administer epinephrine, to provide basic life support if necessary, and to triage the patient to more advanced medical care [10]. Access to vaccination services in pharmacy settings had expanded to all fifty states by 2009 after authorizing pharmacists to administer certain vaccines on a state-by-state basis. During the recent pandemic, the Department of Health and Human Services (HHS) amendment to the Public Readiness and Emergency Preparedness (PREP) Act authorized pharmacy technicians to administer certain immunizations under pharmacist supervision [11]. Pharmacies increase access to immunization services and improve vaccination rates, in part by offering extended hours and convenient locations [12,13]. With the potential for high-risk adverse events, such as anaphylaxis, after vaccine administration, it is crucial for medical professionals in all practice settings to be able to respond and manage these events effectively for patient safety. Currently, it is unknown whether there are differences in death, disability, and hospitalization between various vaccine administration settings. 

With pharmacies being accessible to the public, it is anticipated that the number of vaccination services provided by pharmacy professionals will continue to increase [14]. In the future, more states may expand the scope of practice of pharmacists and incorporate vaccine administration by pharmacy technicians into law as a gateway to increase access to healthcare and improve vaccination rates [15]. However, there is limited literature evaluating outcomes associated with pharmacy immunization services [16]. The aim of this research is to investigate outcomes reported with cases of vaccine anaphylaxis stratified by clinical setting through an analysis of the United States Vaccine Adverse Event Reporting System (VAERS) database from 2017 to 2022. 

## 2. Materials and Methods

### 2.1. Study Design

The VAERS database contains adverse event reports and vaccine error reports that were submitted to the FDA [17]. The database is designed to support the FDA’s post-marketing safety surveillance program for vaccines. The informatic structure of the VAERS database adheres to the safety reporting guidance issued by the International Conference on Harmonisation. Adverse events and medication errors are coded using the Medical Dictionary for Regulatory Activities (MedDRA) terminology [18]. VAERS is a passive reporting system as it relies on individuals to send reports to the Centers for Disease Control and Prevention (CDC) and Food and Drug Administration (FDA). Both the CDC and FDA are public health organizations in the US federal government. Anyone can report an adverse event to VAERS. Additionally, vaccine manufacturers are required to report all adverse events that come to their attention, and healthcare professionals are required to report certain adverse events. Datapoints available within VAERS include patient age, sex, previous allergies, and symptoms; vaccine name, type, manufacturer, lot, dose in series, and location of administration in the patient; administration date, state, and setting (pharmacy, physician’s office, public health clinic, etc.). Outcomes reported in the database include emergency room visits, hospitalizations, disability, and death reported from vaccines. All reports are de-identified, available for query and download via a public dashboard, and do not contain any Health Insurance Portability and Accountability Act (HIPAA) identifiers [17]. Permission is not required to extract data from VAERS [17]. The study was approved by the Binghamton University Institutional Review Board (STUDY00003793, 19 October 2022).

The FDA VAERS database was queried from 1 January, 2017, (when pharmacies were first reported as a vaccine administration setting within the database) to 4 November 2022. The comma-separated value (CSV) files for the study timeframe were downloaded from the public dashboard and converted into an Excel sheet file format [17]. All reports involving anaphylaxis were identified in the Excel sheet file. A report was considered associated with anaphylaxis when the term “anaphylaxis”, “anaphylactic reaction”, “anaphylactic shock” or “anaphylactoid” was coded in the “symptom” field. To be consistent with the typical presentation timeframe, cases of anaphylaxis with an onset within one day of vaccine administration were included, while cases with an onset greater than 1 day were excluded (*n* = 903) [19]. The setting of vaccine administration (i.e., doctor’s office, pharmacy, public health clinic, work setting, nursing home, school or student health clinic, unknown, or other) was noted in the “adminby” field of the report. Death, disability, and hospitalization with each report were noted in the “outcome” fields. Disability was defined in the reporting form as “disability or permanent damage”. Additional characteristics for each report of anaphylaxis included patient age, patient sex, state, date of the event, vaccine type, and history of food or medication allergies.

### 2.2. Statistical Analysis

The primary outcome was the combined prevalence of death or disability when anaphylaxis occurred after vaccine administration for the medical office, pharmacy, and public health clinic setting compared with all other settings. The secondary outcome was the prevalence of hospitalization when anaphylaxis occurred after vaccine administration for the medical office, pharmacy, and public health clinic setting compared with all other settings. Adjusted (age, sex, prior history of allergy, vaccine type) odds ratios (aOR) and associated 95% confidence intervals (CI) were calculated with GraphPad Prism version 9.3.0 for Windows using multiple logistic regression analysis to determine aORs that do not overlap the null value (i.e., OR = 1) at *p* < 0.05 [20]. Age, sex, prior history of allergy, and vaccine type selected for the logistic regression as each variable may impact rates of anaphylaxis [21].

## 3. Results

### 3.1. Descriptive Data

We queried 1,128,235 reports in the VAERS database and identified 2944 cases involving anaphylaxis. As shown in Figure 1, we excluded 903 reports with an onset of the adverse event occurring greater than one day after vaccine administration. Thus, the primary cohort included 2041 cases of anaphylaxis within one day after vaccine administration with representation in the sample from all 50 US states and the District of Columbia. 

The mean age was 43.3 ± 17.5 years, and most cases involved women (1630/2041, 79.9%). Cases of anaphylaxis were reported after receiving a COVID-19 vaccine (1738/2041, 85.2%), influenza vaccine (121/2041, 5.9%), tetanus vaccine (44/2041, 2.2%), zoster vaccine (32/2041, 1.6%), measles vaccine (14/2041, 0.7%), and other vaccines (92/2041, 4.5%). Of the anaphylaxis cases involving COVID-19 vaccines, 1630 (93.8%) were mRNA monovalent vaccines, 14 (0.8%) were mRNA bivalent vaccines, 89 (5.1%) were adenovirus vector vaccines and 5 (0.3%) were unclassified. 

In the primary cohort, 580 anaphylaxis cases (28.4%) occurred in a medical office, 458 (22.4%) in a pharmacy, 140 (6.9%) in a public health clinic, 87 (4.3%) in a workplace setting, 67 (3.3%) in a nursing home, 31 (1.5%) at a student health clinic, 299 (14.7%) at “other” location, and 379 (18.6%) at an “unknown” location (Table 1). A higher proportion of reports from medical offices compared with pharmacies involved persons younger than 18 (13.8% versus 6.6%, respectively), with a history of allergy (64.3% versus 35.6%, respectively), and who received a vaccine other than COVID-19 or influenza (13.9% versus 5.8%, respectively). Likewise, a higher proportion of reports from public health clinics compared with pharmacies involved persons younger than 18 (8.6% versus 6.6%, respectively), with a history of allergy (67.9% versus 35.6%, respectively), and who received a vaccine other than COVID-19 or influenza (7.9% versus 5.8%, respectively).

### 3.2. Outcome Data and Main Results

Reported outcomes included 18 cases (3.1%) of death/disability when anaphylaxis occurred after vaccine administration in the medical provider’s office, 5 cases (1.1%) in the pharmacy setting, and 2 cases (1.4%) in a public health clinic (Table 1). A total of 76 hospitalizations (13.1%) when anaphylaxis occurred after vaccine administration were reported in the medical provider’s office, 24 hospitalizations (5.2%) in the pharmacy setting, and 13 hospitalizations (9.3%) in a public health clinic. In the primary analysis, compared with all other settings, the aOR of death and disability when anaphylaxis occurred was 1.92 (95% CI, 0.86–4.54) in a medical provider’s office, 0.85 (95% CI, 0.26–2.43) in a pharmacy and 1.01 (95% CI, 0.15–3.94) in a public health clinic (Figure 2). 

In the secondary analysis, compared with all other settings, the aOR of hospitalization when anaphylaxis occurred was 1.02 (95% CI, 0.71–1.47) in a medical provider’s office, 1.06 (95% CI, 0.72–1.54) in a pharmacy, and 1.12 (95% CI, 0.61–1.93) in a public health clinic (Figure 3).

## 4. Discussion

### 4.1. Interpretation of Results

Our analysis of outcomes reported with cases of vaccine anaphylaxis stratified by clinical setting revealed non-significant differences in the odds of death/disability and odds of hospitalization between medical offices, pharmacies, and public health clinics compared with all other settings. We utilized six years of adverse event reports from a national database to evaluate outcomes associated with vaccine anaphylaxis by administration setting. We found that cases of anaphylaxis represented 0.3% of all reports in the VAERS database during the study timeframe. A practical finding from our study was that all settings must be prepared to respond to anaphylactic emergencies. Another practical implication from our study includes that cases of death and disability associated with post-vaccination anaphylaxis were rare, which may suggest that healthcare professionals are adequately responding to these medical emergencies.

Most vaccine adverse events following immunization are due to the vaccine stimulating a protective immune response, and not because of allergy. Nonetheless, the serious nature of anaphylactic reactions necessitates that healthcare professionals in all vaccination settings must be trained in the management of this medical emergency [22]. While vaccine anaphylaxis is a frequently studied topic, to our knowledge, this study is the first to evaluate outcomes associated with vaccine anaphylaxis stratified by administration setting [23,24,25]. 

More cases of death and disability were reported when vaccine anaphylaxis occurred in a medical office compared with other settings. However, patients who received immunization services in a medical office had a higher rate of allergy history than patients in other settings, which may indicate that patients at greater risk of adverse events were triaged to a medical office. After adjusting for age, sex, allergy history and vaccine type by regression analysis, the odds of death and disability following vaccine anaphylaxis were not significantly different between settings. 

The rates of hospitalization when vaccine anaphylaxis occurred varied by administration setting from 5.2% of cases in pharmacies to 13.1% in medical offices. This observation may suggest that patients with higher medical complexity or a higher risk of adverse outcomes related to anaphylaxis were seen in medical offices. In the multiple logistic regression analysis, the adjusted odds ratios for hospitalization subsequent to vaccine anaphylaxis were not significantly different by administration setting. 

A tangential finding from our study was that the vast majority of anaphylaxis cases following vaccination involved women (79.9%). In addition to anaphylaxis secondary to vaccines, several studies and reports have observed sex differences such that women experience anaphylaxis induced by food, drugs, and radiocontrast agents at higher rates than men [26,27,28]. Animal models have suggested that anaphylactic reactions are also more severe in females than males [29]. It has been hypothesized that estrogen hormones and X-chromosome-coded factors may regulate the immune response [30]. Our findings are consistent with female vaccine anaphylaxis rates reported in other studies [4,21,31].

Our study identified more reports of anaphylaxis secondary to COVID-19 vaccines than any other vaccine. This may be secondary to mandated reporting and heightened awareness of COVID-19 vaccine safety due to their status under emergency use authorization during the study period [21,32]. The higher number of reported anaphylaxis cases secondary to COVID-19 vaccines also does not reflect the rate of anaphylaxis, as the denominator of total vaccine administration during this time frame is not available via the VAERS database, thus this comparison was not able to be made. As reported previously, COVID-19 vaccines have similar anaphylaxis rates to other immunizations [7].

Another finding in our study was that patients who experienced anaphylaxis after receiving a COVID-19 vaccine had significantly lower odds (0.26, 95% CI 0.11–0.64) of death and disability than patients who had anaphylaxis after receiving other vaccines. This observation may be due to healthcare professionals’ vigilance about safety when administering COVID-19 vaccines. During the timeframe of this study, COVID-19 vaccines emerged as novel therapies using mRNA technology and were made initially available through emergency use authorization, which may have heightened healthcare professionals’ awareness of, reporting of, monitoring for, and readiness to respond to post-vaccination anaphylaxis [21].

This study expands our understanding of the safety of vaccine administration across various settings, including at a time when access to immunization services was expanded during a pandemic [33]. Between December 2020 and September 2022, pharmacists were directly responsible for administering 45% of total COVID-19 vaccines [34]. Pharmacies have also outpaced medical offices in administering influenza vaccines, administering 45.1 million in 2020–21 and 33.8 million in 2021–22 seasons, compared with medical offices administering 28.4 million and 25.4 million, respectively [35]. The results of our study suggest that when anaphylaxis occurs following vaccine administration, healthcare professionals across various settings including pharmacies are responding appropriately. 

The findings of our study may be relevant to public health policies in other countries regarding the administration of vaccines in pharmacies. In many European countries, COVID-19 vaccines cannot be administered by pharmacists or in pharmacies [36]. The role of pharmacists in some European countries is limited to vaccine preparation including reconstitution, or patient education. For example, COVID-19 vaccines cannot be administered in community pharmacies in the Czech Republic, Croatia, Netherlands, Portugal, Spain, or Turkey [36], whereas countries such as the United Kingdom, Ireland and Italy allow COVID-19 vaccines to be administered in community pharmacies [36]. The results of the present study may inform policy makers regarding the safety of vaccine administration by settings and support the direct clinical roles of pharmacists when they have demonstrated training in vaccine administration.

### 4.2. Limitations and Generalizability

One limitation of the VAERS database is that the database is populated with adverse event reports submitted mandatorily and voluntarily from healthcare professionals as well as non-healthcare professionals. Another limitation is that these reports may include incomplete records. Approximately one-third of reports occurred in locations categorized as “other” or “unspecified”, which adds uncertainty to any conclusions regarding administration setting. We suspect the high rate of “other” and “unspecified” locations may be in part due to nontraditional administration settings during the pandemic. Another potential limitation could be the under-reporting of anaphylaxis in VAERS [32]. Death and disability from anaphylaxis after vaccination are rare events, which makes them difficult to quantify accurately [31,32,33,34,35,37]. This study was not designed to compare or report the prevalence of anaphylaxis by vaccine type. The FDA VAERS database reflects vaccine adverse event reports in the United States and may not reflect reporting in other countries. Lastly, cause and effect cannot be determined with observations from surveillance databases [38].

### 4.3. Future Research

Future research utilizing the VAERS database to explore vaccine safety by administration setting might include a disproportionality analysis of specific adverse events following immunization. Another future direction for research might include a focus on reports of vaccination errors by administration setting within the VAERS database. Exploration of additional databases available through the European Medicines Agency and World Health Organization may be insightful for comparison and expand the generalizability of findings. The study of vaccine safety outcomes by administration setting is under-reported, and further research is warranted.

## 5. Conclusions

Anaphylaxis was infrequently reported in the VAERS database. Cases of death and disability associated with post-vaccination anaphylaxis were rare, suggesting that healthcare professionals adequately responded to these emergencies. A multiple regression analysis of a national database across six years revealed non-significant differences in the odds of death/disability and odds of hospitalization associated with post-vaccination anaphylaxis in the medical office, pharmacy, and public health clinic compared with all the other settings. We were reassured to find no signal, suggesting that a particular administration location was associated with significantly higher odds of serious consequences associated with vaccine anaphylaxis. This study expands our understanding of the safety of immunization services by practice settings and confirms that all settings must be prepared to respond to such an emergency.

## Figures and Tables

**Figure 1 vaccines-11-00276-f001:**
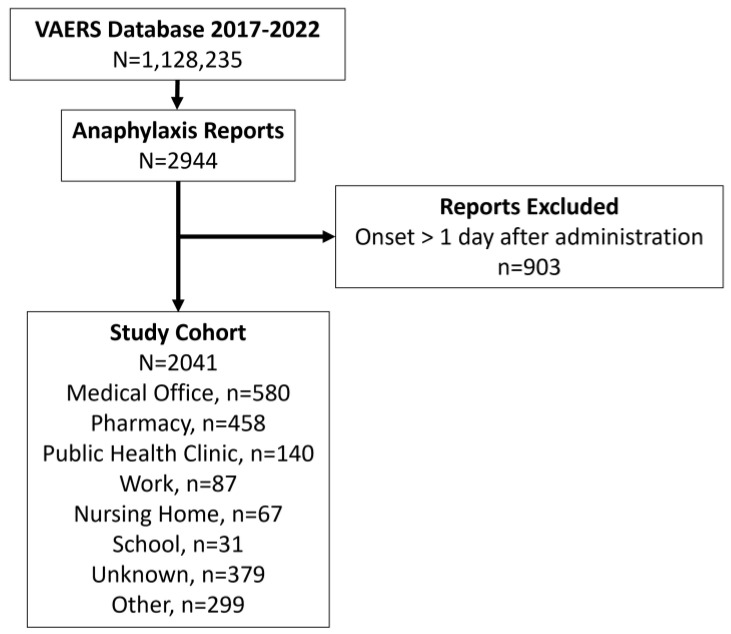
Flowchart of Inclusion Criteria and Cohort Selection.

**Figure 2 vaccines-11-00276-f002:**
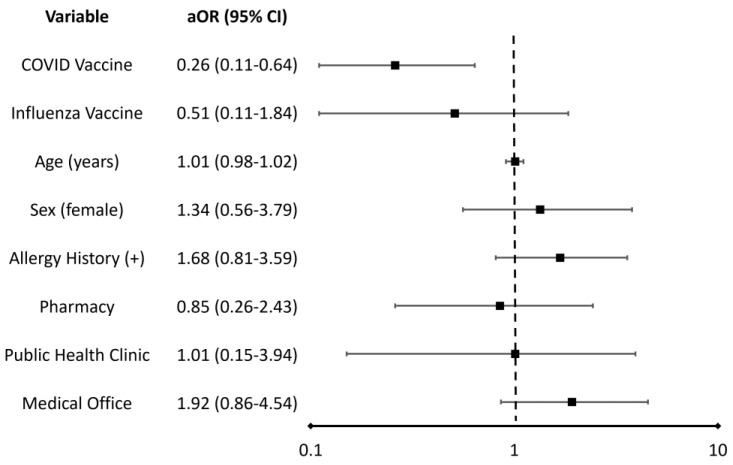
Adjusted odds ratios (aOR) and 95% confidence intervals (CI) of death and disability associated with reports of post-vaccination anaphylaxis for each regression variable *. When holding other covariates constant in the regression analysis, an odds ratio greater than 1 indicates that the outcome of death and disability is more likely to occur when the variable is present. Conversely, an odds ratio less than 1 indicates that the outcome is less likely to occur when the variable is present.

**Figure 3 vaccines-11-00276-f003:**
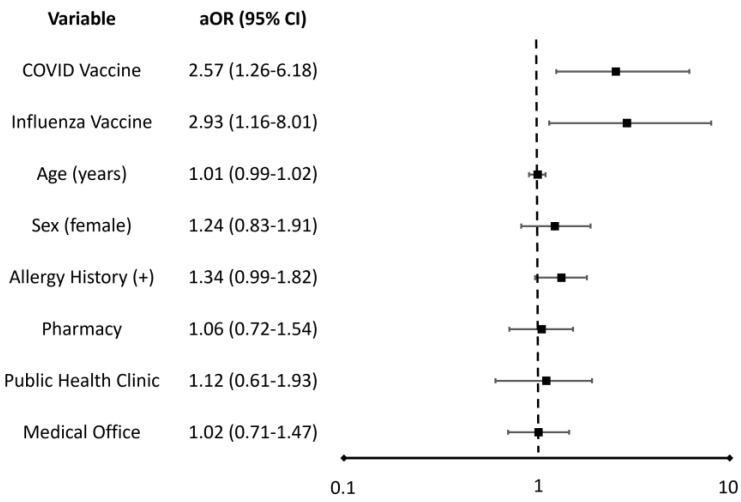
Adjusted odds ratios (aOR) and 95% confidence intervals (CI) of hospitalization associated with reports of post-vaccination anaphylaxis for each regression variable *. When holding other covariates constant in the regression analysis, an odds ratio greater than 1 indicates that the outcome of hospitalization is more likely to occur when the variable is present. Conversely, an odds ratio less than 1 indicates that the outcome is less likely to occur when the variable is present.

**Table 1 vaccines-11-00276-t001:** Demographics and Outcomes of Post-Vaccination Anaphylaxis Reports by Administration Setting.

Variable	Medical Office (*n* = 580)	Pharmacy (*n* = 458)	Public Health Clinic (*n* = 140)	All Other * (*n* = 863)	Total (*n* = 2041)
Sex (female)	471 (81.2%)	356 (77.7%)	116 (82.9%)	687 (79.6%)	1630 (79.9%)
Age (years)	40.4 ± 19.2	43.2 ± 16.4	43.9 ± 17.6	45.4 ± 16.5	43.3 ± 17.5
Age (<18 years)	80 (13.8%)	30 (6.6%)	12 (8.6%)	31 (3.6%)	153 (7.5%)
Allergy History (+)	373 (64.3%)	163 (35.6%)	95 (67.9%)	297 (34.4%)	928 (45.5%)
Vaccine Type: Coronavirus Influenza Tetanus Zoster Other	461 (79.5%) 38 (6.6%) 30 (5.2%) 2 (0.3%) 49 (8.4%)	400 (87.3%) 32 (6.9%) 1 (0.2%) 17 (3.7%) 8 (1.7%)	126 (90.0%) 3 (2.1%) 5 (3.6%) 1 (0.7%) 5 (3.6%)	751 (87.1%) 48 (5.6%) 8 (0.9%) 12 (1.4%) 44 (5.1%)	1738 (85.2%) 121 (5.9%) 44 (2.2%) 32 (1.6%) 106 (5.2%)
Hospitalization	76 (13.1%)	24 (5.2%)	13 (9.3%)	106 (12.3%)	219 (10.7%)
Death & Disability	18 (3.1%)	5 (1.1%)	2 (1.4%)	10 (1.2%)	35 (1.7%)

* Workplace, nursing home or senior living facility, school or student health clinic, “unknown”, and “other” location.

## Data Availability

A publicly available dataset was analyzed in this study. This data can be found here: https://vaers.hhs.gov/about.html (accessed on 5 November 2022).

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
