# Peer review of "Death and Disability Reported with Cases of Vaccine Anaphylaxis Stratified by Administration Setting: An Analysis of the Vaccine Adverse Event Reporting System from 2017 to 2022"

_vaccines, 2023, doi:10.3390/vaccines11020276_

Round 1

Reviewer 1 Report

Even though the field of this article is really important, major revisions must be taken place.

Introductions

Abbreviations should be explained (for example COVID19)

Materials and methods

Please analytically describe excluding criteria in the section 2.1 Study design, because your results section starts with phare: “we excluded 903 reports”.

Moreover, please specify which COVID-19 vaccine was analyzed. It is important to report if different vaccines such as mRNA and Adenovirus vector vaccines had different frequency of reactions.

In your results section, you describe those 31 (1.5%) cases of anaphylaxis was reported from a student health clinic, but unfortunately in the figure 1 "Flowchart» this place of vaccination is not described. Moreover, table 1 contains only 3 places of vaccination separately, and a group of "all others".

Please write clearly "the setting of vaccine administration" in section "Materials and methods", and then report in the text, flowchart, and tables with same manner.

References should be described as follows:

Journal Articles:

1.      Author 1; Author 2. Title of the article. Abbreviated Journal Name Year; Volume: page range.

Moreover, you are required to include in-text references in the main body of your work in square brackets [1].

Furthermore, some references at the end of the list are not numerated.

Conclusions are too short.

Author Response

We are delighted to resubmit our manuscript “Death and disability reported with cases of vaccine anaphylaxis stratified by administration setting: An analysis of the Vaccine Adverse Event Reporting System from 2017 to 2022” for consideration by Vaccines.  We appreciate the thorough and insightful reviewer comments. Major revisions were made to improve clarity of the paper as a whole, reproducibility of the methods, organization of the discussion, and interpretation of the findings. Our responses to the reviewer comments have improved the paper. Each reviewer comment is numbered and addressed below. Revisions and additions are tracked in the manuscript.

Thank you for consideration of this important work. Respectfully.

Reviewer 1

Comments and Suggestions for Authors:

  1. Even though the field of this article is really important, major revisions must be taken place.

Introductions

Abbreviations should be explained (for example COVID19):

Response #1: In the Abstract, Line 29, COVID was replaced with coronavirus. The first time COVID appeared in the body of the manuscript, in the Introduction, first paragraph, line 57, coronavirus was added to explain the abbreviation COVID.

  1. Materials and methods

Please analytically describe excluding criteria in the section 2.1 Study design, because your results section starts with phare: “we excluded 903 reports”.

Response #2: The following sentence in the second paragraph of the methods section was revised.

“To be consistent with the typical presentation timeframe, cases of anaphylaxis with an onset within one day of vaccine administration were included while cases with an onset greater than 1 day were excluded (n=903)[19].”

  1. Moreover, please specify which COVID-19 vaccine was analyzed. It is important to report if different vaccines such as mRNA and Adenovirus vector vaccines had different frequency of reactions.

Response #3: The following sentence was added to the Results, first paragraph.

“Of the anaphylaxis cases involving COVID vaccines, 1,630 (93.8%) were mRNA monovalent vaccines, 14 (0.8%) were mRNA bivalent vaccines, 89 (5.1%) were adenovirus vector vaccines and 5 (0.3%) were unclassified.”

  1. In your results section, you describe those 31 (1.5%) cases of anaphylaxis was reported from a student health clinic, but unfortunately in the figure 1 "Flowchart» this place of vaccination is not described. Moreover, table 1 contains only 3 places of vaccination separately, and a group of "all others".

Please write clearly "the setting of vaccine administration" in section "Materials and methods", and then report in the text, flowchart, and tables with same manner.

Response #4: The following sentences in the Methods section were revised for clarity.

“The setting of vaccine administration (ie, doctor’s office, pharmacy, public health clinic, work setting, nursing home, school or student health clinic, unknown, or other) was noted in the “adminby” field of the report.”

“The primary outcome was the combined prevalence of death or disability when anaphylaxis occurred after vaccine administration for the medical office, pharmacy, and public health clinic setting compared to all other settings. The secondary outcome was the prevalence of hospitalization when anaphylaxis occurred after vaccine administration for the medical office, pharmacy, and public health clinic setting compared to all other settings.”

  1. References should be described as follows:

Journal Articles:

Author 1; Author 2. Title of the article. Abbreviated Journal Name Year; Volume: page range.

Moreover, you are required to include in-text references in the main body of your work in square brackets [1].

Furthermore, some references at the end of the list are not numerated.

Response #5: The citation formatting was reviewed and corrected. Changes were tracked in the manuscript.

  1. Conclusions are too short. 

Response #6: Reviewer 2 suggested that the conclusions should be more concise rather than longer. We attempted to harmonize our response to both reviewer 1 and reviewer 2 regarding the conclusion. The interpretation and discussion of our findings were substantially revised and expanded as documented below in our responses to reviewer 2 and may help to put the conclusions in a better context.

Reviewer 2 Report

This paper is relevant to an international audience, but the paper needs to be improved. Please see my comments about discussion, which is insufficiently developed and structured.

Abstract

-          Please present the full meaning of all abbreviations when first presented in the text (e.g., VAERS).

-          aOR or OR? Is aOR a known abbreviation?

-          “A total of 2,041 cases of anaphylaxis comprised the primary study cohort.” – Number of administered vaccines? Number of countries/regions? Please present more details.

KEYWORDS: please use the maximum number of keywords; please use some MeSH termas. Preferably, use different keywords from those used in the abstract and in the title.

1. Introduction

- “The incidence of anaphylaxis after COVID-19 vaccination has been estimated to be between five and ten cases per million doses [5-7].” Please provide anaphylaxis information per vaccine type? Are there differences mRNA vaccines vs. others vaccine types? Please describe the type of COVID-19 vaccines approved in USA.

- “The serious nature of this adverse event requires healthcare professionals to be adequately trained to respond to post-vaccination hypersensitivity emergencies including anaphylaxis.” References are missing.

- Please briefly explain here What is VAERS? Mission of VAERS? Please present the full meaning of all abbreviations, such as VAERS. Please note that the present paper is to an international audience, and some people do not know What is VAERS?

- Please cite more similar or related papers. For instance, cite some revision papers about anaphylactic reactions of vaccines.

- For instance, see https://pubmed.ncbi.nlm.nih.gov/?term=anaphylactic+vaccine&filter=pubt.meta-analysis&filter=pubt.review&filter=pubt.systematicreview&filter=datesearch.y_10

- New cited references must also be cited in discussion.

Materials and methods

-          Have pharmacovigilance experts being invited to review/read the present paper?

2.1

“VAERS is a passive reporting system as it relies on individuals to send in reports to the Centers for Disease Control and Prevention (CDC) and Food and Drug Administration (FDA).”

-          Please give more details; What is CDC/FDA? Please note that I know What is CDC and FDA, but the paper must be comprehensible to all public/readers.

-          “The study was approved by the Binghamton University Institutional Review Board.” When? Name and composition of the board? Public information website?

-          Regarding “The FDA VAERS database was queried from January 1, 2017, to November 4, 2022, and all reports involving anaphylaxis were extracted. A report was considered associated with anaphylaxis when the term “anaphylaxis”, “anaphylactic reaction”, “anaphylactic shock” or “anaphylactoid” was coded in the “symptom” field. To be consistent with the typical presentation timeframe, cases of anaphylaxis with an onset within one day of vac-cine administration were included and stratified by administration setting [17]. The setting of vaccine administration (ie, doctor’s office, pharmacy, or public health clinic) was noted in the “adminby” field of the report. Death, disability, and hospitalization with each report were noted in the “outcome” fields. Disability was defined in the reporting form as “disability or permanent damage”. Additional characteristics for each report of anaphylaxis included patient age, patient sex, state, date of event, vaccine type, and history of food or medication allergies.” Please present a flowchart. How data were reviewed/rechecked? How and where data were stored? Please give more details about data collection procedure, analysis, and review. All studies must be reproductible. Please present the link to the database VAERS.

-          Please cite and apply the Equator guidelines https://www.equator-network.org/, which will increase the quality of the paper.

2.2. Statistical Analysis

Who conducted the statical analysis? Were data rechecked? Validation procedures?

3. Results

-          Results are too compact; please use some subheadings. For instance, present data according to the definition of outcomes 1 and 2 “The primary outcome was the combined prevalence of death or disability for each setting while the secondary outcome was the prevalence of hospitalization.”

-          Please better explain the findings of Figure 2 and 3. Please note that this type of representation is not familiar to all readers. Please explain the difference between the values of OR higher and lower than 1. Why medical office “1.92 (0.86-4.54)” is not a confounding variable (Figure 2)? Why people in medical offices died almost two more times than in other places? This finding is strange… Or not?

4. Discussion

- Please cite the new introduced/cited references in introduction (revision studies about similar or related topics).

- Please explain all the findings in discussion. Please explain/discuss all the findings from Table 1 in discussion. Please explain/discuss all the findings from Figure 1 in discussion. Please explain/discuss all the findings from Figure 2 in discussion. Please check one by one if all presented findings are explained in discussion.

- Please do not start discussion by explaining the data from Figure 3. Please organize discussion according to the presentation of data in the section of results.

- I also recommend the use of some subheadings to organize discussion, although subheadings are facultative in discussion. For instance, present data according to the definition of outcomes 1 and 2 “The primary outcome was the combined prevalence of death or disability for each setting while the secondary outcome was the prevalence of hospitalization.”

- “We found that cases of anaphylaxis represented 0.3% of all reports in the VAERS database during the study timeframe.” Please present data from other databases. For instance, see the databases of WHO and EMA, which are free and public. You need to compare the present data with data from other regions/countries.

- “However, patients with an allergy history had a higher odds (1.68, 95% CI 0.81-3.59), although nonsignificant, of death and disability following vaccine anaphylaxis.” This sentence is not clear. Please give more details. Preferably, do not present findings in discussion.

- “It has been hypothesized that estrogen hormones and X-chromosome coded factors may regulate the immune response [29].” Is this the only possible explanation? What about the prevalence of immunologic reactions?

- In my opinion, the discussion is the worst section of the present paper. Discussion is insufficiently developed and organized/structured. The rational of discussion is not well structured. Sincerely, I highly recommend a reconstruction of study discussion. Please read more papers and deeply investigate the possible causality explanations. You need to study more papers per each variable. Please study/read more papers! Ok?

- “This study was not designed to compare or report the prevalence of ana-phylaxis by vaccine type.” This information should be presented and explained in methods. Is this a possible study limitation?

- Please present three subheadings at the end of Discussion: study limitations, practical implications, and future research.

Conclusion

-          The conclusion should reply to study objective and to outcomes 1 and 2.

-          Other information should be placed in Discussion.

Figures and Tables

-          Please check the format of Figures and Tables in instructions for authors.

-          Please see published papers from vaccines.

References

-          Please cite more references, such as more review studies.

-          Please cite more ADRs databases. For instance, see the EMA, WHO and TGA databases of ADRs, which are free.

-          Please check the format of all references in instructions for authors.

Merry Christmas and Happy New Year 2023! 

Author Response

We are delighted to resubmit our manuscript “Death and disability reported with cases of vaccine anaphylaxis stratified by administration setting: An analysis of the Vaccine Adverse Event Reporting System from 2017 to 2022” for consideration by Vaccines.  We appreciate the thorough and insightful reviewer comments. Major revisions were made to improve clarity of the paper as a whole, reproducibility of the methods, organization of the discussion, and interpretation of the findings. Our responses to the reviewer comments have improved the paper. Each reviewer comment is numbered and addressed below. Revisions and additions are tracked in the manuscript.

Thank you for consideration of this important work. Respectfully.

Reviewer 2

Comments and Suggestions for Authors:

  1. This paper is relevant to an international audience, but the paper needs to be improved. Please see my comments about discussion, which is insufficiently developed and structured.

Abstract

Please present the full meaning of all abbreviations when first presented in the text (e.g., VAERS).

Response #1: The abstract was revised on line 22 in order to define the VAERS abbreviation.

In the Abstract, Line 29, COVID was replaced with coronavirus. The first time COVID appeared in the body of the manuscript in the Introduction, first paragraph, line 57, coronavirus was added to explain the abbreviation COVID. The VAERS abbreviation was defined the first time is was listed in the manuscript.

  1. aOR or OR? Is aOR a known abbreviation?

Response #2: “aOR” is defined in the abstract in the following sentence.

“Adjusted (age, sex, prior history of allergy, vaccine type) odds ratios (aOR) and associated 95% confidence intervals (CI) were calculated using logistic regression analysis.”

The rationale for variables selected for the aOR was clarified in the last sentence of the methods section and a citation was added to support the use of these variables.

  1. “A total of 2,041 cases of anaphylaxis comprised the primary study cohort.” – Number of administered vaccines? Number of countries/regions? Please present more details.

Response #3: The sentence in the abstract was expanded to include:

“A total of 2,041 cases of anaphylaxis comprised the primary study cohort with representation in the sample from all 50 US states and the District of Columbia.”

  1. KEYWORDS: please use the maximum number of keywords; please use some MeSH termas. Preferably, use different keywords from those used in the abstract and in the title.

Response #4: COVID-19 vaccination and influenza vaccination were added as keywords.

  1. Introduction

“The incidence of anaphylaxis after COVID-19 vaccination has been estimated to be between five and ten cases per million doses [5-7].” Please provide anaphylaxis information per vaccine type? Are there differences mRNA vaccines vs. others vaccine types? Please describe the type of COVID-19 vaccines approved in USA.

Response #5: The following sentence and an additional citation was added which references information for both mRNA and adenovirus vaccines.

Excipients in both the mRNA and adenovirus vector-based COVID-19 vaccines have been found to be allergenic [8].

  1. “The serious nature of this adverse event requires healthcare professionals to be adequately trained to respond to post-vaccination hypersensitivity emergencies including anaphylaxis.” References are missing.

Response #6: The following reference was cited.

Kim MA, Lee YW, Kim SR, et al. COVID-19 Vaccine-associated Anaphylaxis and Allergic Reactions: Consensus Statements of the KAAACI Urticaria/Angioedema/Anaphylaxis Working Group. Allergy Asthma Immunol Res. 2021;13(4):526-544.

  1. Please briefly explain here What is VAERS? Mission of VAERS? Please present the full meaning of all abbreviations, such as VAERS. Please note that the present paper is to an international audience, and some people do not know What is VAERS?

Response #7: The explanation of VAERS was expanded in the Methods section of the manuscript. The citation of the public access to the VAERS database was advanced to the first sentence of the Methods section in order to make the reference more prominent.

  1. Please cite more similar or related papers. For instance, cite some revision papers about anaphylactic reactions of vaccines.

For instance, see https://pubmed.ncbi.nlm.nih.gov/?term=anaphylactic+vaccine&filter=pubt.meta-analysis&filter=pubt.review&filter=pubt.systematicreview&filter=datesearch.y_10

New cited references must also be cited in discussion.

Response #8: Additional references were added to the introduction and discussion. The reference numbering was updated. The bibliography was updated accordingly.

  1. Materials and methods

Have pharmacovigilance experts being invited to review/read the present paper?

Response #9: In addition to the expertise of the authors of this paper, we consulted with Dr. Yuan Fang. Dr. Fang is a PhD trained biostatistician who has completed a postdoctoral fellowship at the Boston University School of Public Health. Dr. Fang validated our methodological approach to the study.

  1. “VAERS is a passive reporting system as it relies on individuals to send in reports to the Centers for Disease Control and Prevention (CDC) and Food and Drug Administration (FDA).”

Please give more details; What is CDC/FDA? Please note that I know What is CDC and FDA, but the paper must be comprehensible to all public/readers.

Response #10: The following sentence was added in the Methods, line 95.

Both the CDC and FDA are public health organizations in the US federal government.

  1. “The study was approved by the Binghamton University Institutional Review Board.” When? Name and composition of the board? Public information website?

Response #11: The sentence was expanded to include the following:

The study was approved by the Binghamton University Institutional Review Board (STUDY00003793, October 19, 2022).

  1. Regarding “The FDA VAERS database was queried from January 1, 2017, to November 4, 2022, and all reports involving anaphylaxis were extracted. A report was considered associated with anaphylaxis when the term “anaphylaxis”, “anaphylactic reaction”, “anaphylactic shock” or “anaphylactoid” was coded in the “symptom” field. To be consistent with the typical presentation timeframe, cases of anaphylaxis with an onset within one day of vac-cine administration were included and stratified by administration setting [17]. The setting of vaccine administration (ie, doctor’s office, pharmacy, or public health clinic) was noted in the “adminby” field of the report. Death, disability, and hospitalization with each report were noted in the “outcome” fields. Disability was defined in the reporting form as “disability or permanent damage”. Additional characteristics for each report of anaphylaxis included patient age, patient sex, state, date of event, vaccine type, and history of food or medication allergies.” Please present a flowchart. How data were reviewed/rechecked? How and where data were stored? Please give more details about data collection procedure, analysis, and review. All studies must be reproductible. Please present the link to the database VAERS.

Response #12: The citation of the public access to the VAERS database was advanced to the first sentence of the Methods section in order to make the reference more prominent. The Methods section was expanded to describe additional data steps and the file format.

Citation: U.S. Department of Health and Human Services. Vaccine Adverse Event Reporting System (VAERS). Accessed October 16, 2022. Available at: https://vaers.hhs.gov/about.html  

  1. Please cite and apply the Equator guidelines https://www.equator-network.org/, which will increase the quality of the paper.

Response #13: The STROBE checklist as documented in the following citation was used to improve the reporting of key elements in the study design.

von Elm E, Altman DG, Egger M, et al. The Strengthening the Reporting of Observational Studies in Epidemiology (STROBE) statement: guidelines for reporting observational studies. Lancet. 2007;370(9596):1453-1457.

  1. Statistical Analysis

Who conducted the statical analysis? Were data rechecked? Validation procedures?

Response #14: The inferential analysis was conducted by Professor Kenneth McCall and validated by Professor Rachel Klosko. In addition to the expertise of the authors of this paper, we consulted with Dr. Yuan Fang. Dr. Fang is a PhD trained biostatistician who has completed a postdoctoral fellowship at the Boston University School of Public Health. Dr. Fang validated our methodological approach to the study.

  1. Results

Results are too compact; please use some subheadings. For instance, present data according to the definition of outcomes 1 and 2 “The primary outcome was the combined prevalence of death or disability for each setting while the secondary outcome was the prevalence of hospitalization.”

Response #15: The results section was divided into two subheadings (3.1 Descriptive Data and 3.2 Outcome Data and Main Results) for improved clarity.

  1. Please better explain the findings of Figure 2 and 3. Please note that this type of representation is not familiar to all readers. Please explain the difference between the values of OR higher and lower than 1. Why medical office “1.92 (0.86-4.54)” is not a confounding variable (Figure 2)? Why people in medical offices died almost two more times than in other places? This finding is strange… Or not?

Response #16: When holding other covariates constant in the regression analysis as illustrated in Figure 2, an odds ratio greater than 1 indicates that the outcome of death and disability is more likely to occur when the variable is present and conversely an odds ratio less than 1 indicates that the outcome is less likely to occur when the variable is present. As shown in Figure 3, an odds ratio greater than 1 indicates that the outcome of hospitalization is more likely to occur when the variable is present and conversely an odds ratio less than 1 indicates that the outcome is less likely to occur when the variable is present.

This explanation was added as a footnote to each figure.

The interpreted in the discussion of the finding regarding the outcome in the medical office setting was expanded.

  1. Discussion

Please cite the new introduced/cited references in introduction (revision studies about similar or related topics).

Response #17: Additional references were added to the introduction and discussion. The reference numbering and bibliography were updated accordingly.

  1. Please explain all the findings in discussion. Please explain/discuss all the findings from Table 1 in discussion. Please explain/discuss all the findings from Figure 1 in discussion. Please explain/discuss all the findings from Figure 2 in discussion. Please check one by one if all presented findings are explained in discussion.

Response #18: Major revisions were added to the discussion including the addition of subheadings to improve organization and an expanded explanation/interpretation of the findings.

  1. Please do not start discussion by explaining the data from Figure 3. Please organize discussion according to the presentation of data in the section of results.

Response #19: We chose to begin the discussion by summarizing key results based on the primary study objectives as recommended in the discussion format of the STROBE guidelines.

  1. I also recommend the use of some subheadings to organize discussion, although subheadings are facultative in discussion. For instance, present data according to the definition of outcomes 1 and 2 “The primary outcome was the combined prevalence of death or disability for each setting while the secondary outcome was the prevalence of hospitalization.”

Response #20: The discussion was divided into subheadings (interpretation of results, limitations and generalizability, and future research) as recommended in the STROBE guidelines.

  1. “We found that cases of anaphylaxis represented 0.3% of all reports in the VAERS database during the study timeframe.” Please present data from other databases. For instance, see the databases of WHO and EMA, which are free and public. You need to compare the present data with data from other regions/countries.

Response #21: Exploration of additional databases available through the European Medicines Agency and World Health Organization were suggested as future areas of research within the discussion.

  1. “However, patients with an allergy history had a higher odds (1.68, 95% CI 0.81-3.59), although nonsignificant, of death and disability following vaccine anaphylaxis.” This sentence is not clear. Please give more details. Preferably, do not present findings in discussion.

Response #22: This sentence was deleted.

  1. “It has been hypothesized that estrogen hormones and X-chromosome coded factors may regulate the immune response [29].” Is this the only possible explanation? What about the prevalence of immunologic reactions?

Response #23: This paragraph was revised and expanded with additional citations.

  1. In my opinion, the discussion is the worst section of the present paper. Discussion is insufficiently developed and organized/structured. The rational of discussion is not well structured. Sincerely, I highly recommend a reconstruction of study discussion. Please read more papers and deeply investigate the possible causality explanations. You need to study more papers per each variable. Please study/read more papers! Ok?

Response #24: Thank you for this important critique. We have made major revisions to the discussion including expanded explanation of the results and organization with subheadings based on the STROBE guidelines.

  1. “This study was not designed to compare or report the prevalence of ana-phylaxis by vaccine type.” This information should be presented and explained in methods. Is this a possible study limitation?

Please present three subheadings at the end of Discussion: study limitations, practical implications, and future research.

Response #25: Subheadings (interpretation of results, limitations and generalizability, and future research) were added to the discussion as recommended and based on STROBE guidelines. A new section in the discussion on future research was developed.

  1. Conclusion

The conclusion should reply to study objective and to outcomes 1 and 2.

Other information should be placed in Discussion.

Response #26: Reviewer #1 commented that “Conclusions are too short.” We attempted to harmonize our response to both reviewer 1 and reviewer 2 regarding the conclusion. The interpretation and discussion of our findings were substantially revised and expanded as documented above in our responses and may help to put the conclusions in a better context.

  1. Figures and Tables

Please check the format of Figures and Tables in instructions for authors.

Please see published papers from vaccines.

Response #27: Table and figure formatting was checked, and revisions were made. All revisions were tracked in the manuscript.

  1. References

Please cite more references, such as more review studies.

Please cite more ADRs databases. For instance, see the EMA, WHO and TGA databases of ADRs, which are free.

Please check the format of all references in instructions for authors.

Response #28: Additional references were added to the introduction and discussion. The reference numbering and bibliography were updated. Additional ADR databases such as EMA and WHO were added to the discussion.

Reviewer 3 Report

Thank you for the invitation. I have read the manuscript with great interest and found it timely and important during the era of vaccination. I have a few concerns regarding manuscript.

The authors are encouraged to provide more information in the VAERS, as most of the researchers and healthcare practitioners are not very familiar with CDC/FDAs` reporting system. Please also indicate in the manuscript section that permission to use the data is not required.  Please also provide the ethical approval number and the name of body approving the protocol of this study.

The authors are requested to provide clear definitions of the primary and secondary outcomes of this study. Moreover, is it possible to provide the data as a supplementary file? I assumed that the authors have collected all the data from 2017 to 2022 and only used data for which hypersensitivity reactions were reported. This data will help the readers to familiarize themselves with the types of variables included in VAERS data.

The process of variables selection for logistic regression is not clear. How were the variables selected to put in the logistic regression and what variables from the univariate model were subjected to the multivariate analysis?

Can the authors be more specific about the other vaccines? I know that it is sometimes reported as other in the VAERS reporting system. In this context, it is advisable to include the types of variables used in this study in the method section. The method section should clearly illustrate the available variables in VAERS records, explanation and definition of each variable.

Reported outcomes included 18 cases (3.1%) of death/disability in the medical pro- 139 vider’s office, 5 cases (1.1%) of death/disability in the pharmacy setting, and 2 cases (1.4%) 140 of death/disability in a public health clinic (Table 1). if I am not wrong, the authors are stating that 5 cases who took vaccine from pharmacy experienced death/disability AND 2 cases who took vaccine from public health clinical experienced death/disability. This clarification is needed in the method, results as well as discussion section.

Line 183-189: The authors have well described that the female experiences more reactions than males but close this discussion with a simple and vague argument. A similar pattern of discussion was observed at various points in this section. The authors are encouraged to open the discussion and close the sentence in a conclusive way, particularly where the authors are discussing the risk factors.

This study reported more anaphylactic reactions following the administration of COVID-19 vaccines. How these findings will impact the vaccination coverage, as many people may become afraid if they become aware about the deaths and disabilities due to COVID-19 vaccines. In the era of vaccine hesitancy, the authors are encouraged to provide a passage on this issue.

The pharmacists` involvement in the vaccination campaign should be discussed in one paragraph in the discussion section. The importance of pharmacist in vaccination programs has been well acknowledged during the pandemic due to easily availability, economy and accessibility of the pharmacist for the public. I would suggest the authors consider a paragraph at the end of the discussion and close this paragraph by underscoring the importance of pharmacists in public health measures.

Author Response

We are delighted to resubmit our manuscript “Death and disability reported with cases of vaccine anaphylaxis stratified by administration setting: An analysis of the Vaccine Adverse Event Reporting System from 2017 to 2022” for consideration by Vaccines.  We appreciate the thorough and insightful reviewer comments. Major revisions were made to improve clarity of the paper as a whole, reproducibility of the methods, organization of the discussion, and interpretation of the findings. Our responses to the reviewer comments have improved the paper. Each reviewer comment is numbered and addressed below. Revisions and additions are tracked in the manuscript.

Thank you for consideration of this important work. Respectfully.

Reviewer 3

  1. The authors are encouraged to provide more information in the VAERS, as most of the researchers and healthcare practitioners are not very familiar with CDC/FDAs` reporting system. Please also indicate in the manuscript section that permission to use the data is not required.  Please also provide the ethical approval number and the name of body approving the protocol of this study.

Response #1: The citation of the public access to the VAERS database was advanced to the first sentence of the Methods section in order to make the reference more prominent.

Statement on public availability of data added to line 113. IRB approval number and approval date added to line 114.

  1. The authors are requested to provide clear definitions of the primary and secondary outcomes of this study. Moreover, is it possible to provide the data as a supplementary file? I assumed that the authors have collected all the data from 2017 to 2022 and only used data for which hypersensitivity reactions were reported. This data will help the readers to familiarize themselves with the types of variables included in VAERS data.

Response #2: Primary and secondary endpoint were further clarified in the methods section in lines 134 through 138.

The variables available in the VAERS database were additionally listed in the methods section in lines 106-110.

The Data Availability Statement was confirmed: “A publicly available dataset was analyzed in this study. This data can be found here: https://vaers.hhs.gov/about.html”

  1. The process of variables selection for logistic regression is not clear. How were the variables selected to put in the logistic regression and what variables from the univariate model were subjected to the multivariate analysis?

Response #3: The following sentence and an additional citation were added to the last sentence of the methods section to support the rationale for selecting variables for the regression. “Age, sex, prior history of allergy, and vaccine type were selected for the logistic regression as each variable may impact rates of anaphylaxis [21].”

  1. Can the authors be more specific about the other vaccines? I know that it is sometimes reported as other in the VAERS reporting system. In this context, it is advisable to include the types of variables used in this study in the method section. The method section should clearly illustrate the available variables in VAERS records, explanation and definition of each variable.

Response #4: Vaccine types classified as “Other” in this study, were ones that were not classified into any of the other listed categories of vaccine including, COVID-19 vaccines, influenza vaccines, tetanus vaccines, zoster vaccines, and measles vaccines. Variables available in the VAERS databased that were assessed in this study are included in lines 106-110.

  1. Reported outcomes included 18 cases (3.1%) of death/disability in the medical provider’s office, 5 cases (1.1%) of death/disability in the pharmacy setting, and 2 cases (1.4%) 140 of death/disability in a public health clinic (Table 1). if I am not wrong, the authors are stating that 5 cases who took vaccine from pharmacy experienced death/disability AND 2 cases who took vaccine from public health clinical experienced death/disability. This clarification is needed in the method, results as well as discussion section.

Response #5: Clarification to this statement was provided in lines 126-128 of the methods section, lines 181-185 of the results section, and 218-220 of the discussion section.

  1. Line 183-189: The authors have well described that the female experiences more reactions than males but close this discussion with a simple and vague argument. A similar pattern of discussion was observed at various points in this section. The authors are encouraged to open the discussion and close the sentence in a conclusive way, particularly where the authors are discussing the risk factors.

Response #6: Thank you for this critique. Two new statements and citations were added to this paragraph regarding this point. Our study findings are consistent with other studies looking at vaccine anaphylaxis rates based on sex.

  1. This study reported more anaphylactic reactions following the administration of COVID-19 vaccines. How these findings will impact the vaccination coverage, as many people may become afraid if they become aware about the deaths and disabilities due to COVID-19 vaccines. In the era of vaccine hesitancy, the authors are encouraged to provide a passage on this issue.

Response #7: Thank you. This is an important clarification point for this paper because we do not want to portray the message that there is significantly more anaphylaxis to the COVID-19 vaccine, when we don’t have a denominator to compare all other vaccines to assess this rate. A paragraph was added in the discussion (lines 254-261) to ensure that this point is clear. Please advise if statement requires further clarification for readers to ensure appropriate understanding.

  1. The pharmacists` involvement in the vaccination campaign should be discussed in one paragraph in the discussion section. The importance of pharmacist in vaccination programs has been well acknowledged during the pandemic due to easily availability, economy and accessibility of the pharmacist for the public. I would suggest the authors consider a paragraph at the end of the discussion and close this paragraph by underscoring the importance of pharmacists in public health measures.

Response #8: We appreciate this comment. A paragraph was added in the discussion to lines 271-279 to highlight this point and the importance of the pharmacist’s role in this area of public health.

Reviewer 4 Report

The introduction part is extremely well written. Authors have comprehensively explained the importance of vaccines and the risk of anaphylaxis reactions. However authors should explain why they have stratified risk according to clinical settings in current review. Moreover, previous relevant literature especially systematic review/meta analysis should be mentioned clearly in the introduction section. There is no reference of previously conducted studies on this association in the introduction section.

Authors should explain the reason why the database was searched from 2017 to 2022, is there any specific reason authors did not include previous years as basically any retrospective data inspection should be atleast for 10 years.

The discussion section should add more information regarding anaphylaxis with respect to various vaccines with possible reasons. Authors have focused more on study settings that is understandable as this is their objective, but type of vaccines and anaphylaxis reactions should be explained in detail.

Author Response

We are delighted to resubmit our manuscript “Death and disability reported with cases of vaccine anaphylaxis stratified by administration setting: An analysis of the Vaccine Adverse Event Reporting System from 2017 to 2022” for consideration by Vaccines.  We appreciate the thorough and insightful reviewer comments. Major revisions were made to improve clarity of the paper as a whole, reproducibility of the methods, organization of the discussion, and interpretation of the findings. Our responses to the reviewer comments have improved the paper. Each reviewer comment is numbered and addressed below. Revisions and additions are tracked in the manuscript.

Thank you for consideration of this important work. Respectfully.

Reviewer 4

  1. The introduction part is extremely well written. Authors have comprehensively explained the importance of vaccines and the risk of anaphylaxis reactions. However, authors should explain why they have stratified risk according to clinical settings in current review. Moreover, previous relevant literature especially systematic review/meta-analysis should be mentioned clearly in the introduction section. There is no reference of previously conducted studies on this association in the introduction section.

Response #1: The authors have stratified risk according to practice setting as it has not been described in previous literature. The authors felt that this was an important question to answer, with the majority of vaccines now administered outside of a medical providers office. It is important to know that medical professionals in other practice settings are able to manage anaphylaxis events and safely administer vaccines to patients. I few sentences were included in the introduction to highlight this risk (lines 78-82) but to our knowledge, there is no other meta-analysis/systematic review data that exists on this topic, hence the need for this paper.

  1. Authors should explain the reason why the database was searched from 2017 to 2022, is there any specific reason authors did not include previous years as basically any retrospective data inspection should be at least for 10 years

Response #2: Thank you for asking for this clarification. A statement was added to lines 116-117 in the methods section. Pharmacies were first included as a vaccine administration setting in the VAERS database starting in 2017, which is why we included data starting at that date. Prior to 2017, we do not have VAERS information regarding vaccine administration in pharmacies.

  1. The discussion section should add more information regarding anaphylaxis with respect to various vaccines with possible reasons. Authors have focused more on study settings that is understandable as this is their objective, but type of vaccines and anaphylaxis reactions should be explained in detail.

Response #3: This was expanded in the results section in lines 161-163 and in the discussion section in lines 254-261. We emphasized that the rates of anaphylaxis after COVID-19 vaccination are not significantly higher than other vaccines, despite more frequent reporting of anaphylaxis compared to other vaccines. An additional citation was added to support this interpretation. This can be elaborated on in more detail if necessary. However, with most of our data involving COVID-19 and influenza vaccines, the authors felt this was sufficient.

Round 2

Reviewer 1 Report

All my corrections and recommendations were added.

Author Response

Your review of our paper is appreciated.

Reviewer 2 Report

Dear authors, Thanks for all corrections and updates.

Minor comments:

- Please comment the fact that COVID-19 vaccines cannot be administered in pharmacies in many countries, including countries where community pharmacies and pharmaceuticals are prepared to administered vaccines (e.g. flu vaccines) and how the findings from the present study can be relevant, regarding the public health policies of these countries. For instance, in Portugal community pharmacies cannot administer the COVID vaccines, but can administer flu vaccines or other injectables.... The same occur in other countries from the European Union...

Data from European Union (administration places of COVID-19 vaccines) can be consulted here

Paudyal, V., Fialová, D., Henman, M.C. et al. Pharmacists’ involvement in COVID-19 vaccination across Europe: a situational analysis of current practice and policy. Int J Clin Pharm 43, 1139–1148 (2021). https://doi.org/10.1007/s11096-021-01301-7

Best regards,

Author Response

Your review of our paper is appreciated. Thank you for suggesting that we address how our research may be relevant to public health policies in other counties. We have added a paragraph to the discussion that addresses this point. We have also cited the article by Paudyal and colleagues.